Characterizing the adult and larval transcriptome of the multicolored Asian lady beetle, Harmonia axyridis

Havens Lindsay A. havens.lindsay2390@gmail.com lao95@wildcats.unh.edu
MacManes Matthew D.
Department of Molecular, Cellular, and Biomedical Sciences, University of New Hampshire , Durham , NH , United States
Emes Richard
Electronic publication date: 2016 Jun 14
Publication date: 2016
Volume: 4
Electronic Location ID: e2098
Received 2015 Dec 16; Accepted 2016 May 10
Copyright: ©2016 Havens and MacManes
Copyright year: 2016
Copyright holder: Havens and MacManes
License: This is an open access article distributed under the terms of the Creative Commons Attribution License, which permits unrestricted use, distribution, reproduction and adaptation in any medium and for any purpose provided that it is properly attributed. For attribution, the original author(s), title, publication source (PeerJ) and either DOI or URL of the article must be cited.
License URL: https://creativecommons.org/licenses/by/4.0/

Keywords: Harmonia axyridis, Ladybeetle, Transcriptome

Funding: University of New Hampshire This project was supported by MacManes lab startup funds provided by the University of New Hampshire. The funders had no role in study design, data collection and analysis, decision to publish, or preparation of the manuscript.

==============================
The reasons for the evolution and maintenance of striking visual phenotypes are as widespread as the species that display these phenotypes. While study systems such as Heliconius and Dendrobatidae have been well characterized and provide critical information about the evolution of these traits, a breadth of new study systems, in which the phenotype of interest can be easily manipulated and quantified, are essential for gaining a more general understanding of these specific evolutionary processes. One such model is the multicolored Asian lady beetle, Harmonia axyridis, which displays significant elytral spot and color polymorphism. Using transcriptome data from two life stages, adult and larva, we characterize the transcriptome, thereby laying a foundation for further analysis and identification of the genes responsible for the continual maintenance of spot variation in H. axyridis.

Introduction

The evolution and maintenance of phenotypic polymorphism and striking visual phenotypes have fascinated scientists for many years (Darwin, 1859; Endler, 1986; Fisher, 1930; Gray & McKinnon, 2007; Joron et al., 2006). In general, insects have become increasingly popular as study organisms to examine phenotypic variation (Jennings, 2011; Joron et al., 2006). One such insect displaying extensive elytra and spot variation that has yet to be extensively studied is the Asian Multicolored Ladybeetle, Harmonia axyridis.

The mechanisms responsible for the evolution of these phenotypes are as widespread as the species that display them. Aposematism, crypsis, and mimicry may play a role in the evolution of phenotypic variation in the animal kingdom. Members of family Dendrobatidae, poison dart frogs, are aposematically colored (Cadwell, 1996), while Tetrix subulata grasshoppers maintain their phenotypic polymorphism to aid in crypsis (Karpestam, Merilaita & Forsman, 2014). A mimicry strategy is utilized by one particularly well-characterized species that exhibits phenotypic polymorphism, the Neotropic butterfly system, Heliconius. The color, pattern, and eyespot polymorphism seen in Heliconius is thought to have arose as a result of Müllerian mimicry (Flanagan et al., 2004) and the supergenes underlying these traits have been well characterized (Kronforst et al., 2006; Joron et al., 2006; Jones et al., 2012).

These studies, aiming to elucidate the mechanistic links between phenotype and genotype, present a unique opportunity to gain insight into the inner workings of many important evolutionary processes. While systems like poison frogs and butterflies have been pioneering, the use of novel models, especially those that can be easily manipulated, are needed. One such study system that possesses many of the benefits of classical models, while offering several key benefits, described below, is the multicolored Asian lady beetle, Harmonia axyridis. Harmonia, which is common throughout North America, and easily bred in laboratory environments, possesses significant variation in elytral spot number and color.

Elytra color can be red, orange, yellow, or black and spot numbers of H. axyridis range from zero to twenty-two (L Havens, pers. obs., 2013). The patterning is symmetrical on both wings. In some animals, there is a center spot beneath the pronotum which leads to an odd number of spots. The elytral spots are formed by the production of melanin pigments (Bezzerides et al., 2007). The frequency of different morphs varies with location and temperature (Michie et al., 2010). The melanic morph is more prevalent in Asia when compared to North America (LaMana & Miller, 1996; Dobzhansky, 1993). A decrease in melanic H. axyridis has been shown to be correlated with an increase in average yearly temperatures in the Netherlands (Brakefield & De Jong, 2011).

Sexual selection may play a role in color variation in H. axyridis. Osawa & Nishida (1992) remarked that female H. axyridis might choose their mates based on melanin concentration. Their choice, however, has been shown to vary based on season and temperature. Non-melanic (red, orange, or yellow with any spot number) males have a higher frequency of mating in the spring-time, while melanic (black) males have an increased frequency of mating in the summer. While this has been shown with respect to elytral color, no such findings have occurred for spot number. Although these spot patterns are believed to be related to predator avoidance, thermotolerance, or mate choice (Osawa & Nishida, 1992), the genetics underlying these patterns is currently unknown.

To begin to understand the genomics of elytral coloration and spot patterning, we sequenced the transcriptome of a late-stage larva and adult ladybug. These results lay the groundwork for future study of the genomic architecture of pigment placement and development in H. axyridis.

Methods and Materials

Specimen capture, RNA extraction, library prep and sequencing

One larval (Fig. 1A) and one adult (Fig. 1B) H. axyridis were captured on the University of New Hampshire campus in Durham, New Hampshire (43.1339°N, 70.9264°W). The adult was orange with 18 spots. The insects were placed in RNAlater and immediately stored in a −80C freezer until RNA extraction was performed. The RNA from both individuals was extracted following the TRIzol extraction protocol (Invitrogen, Carlsbad, CA, USA). The entire insect was used for the RNA extraction protocol. The quantity and quality of extracted RNA was analyzed using a Qubit (Life Technology, Carlsbad USA) as well as a Tapestation 2200 (Agilent technologies, Palo Alto, CA, USA) prior to library construction. Following verification, RNA libraries were constructed for both samples following the TruSeq stranded RNA prep kit (Illumina, San Diego, CA, USA), which includes PolyA purification of mRNA. To allow multiple samples to be run in one lane, a unique index code was added to each sample. These samples were then pooled in equimolar quantities using Tapestation analysis. The multiplexed libraries were then sent to the New York Genome Center (New York, NY, USA) for sequencing on a single lane (125 bp paired end) of the HiSeq 2500 sequencer.

Figure 1 (A) The larva used for transcriptome sequencing. (B) The adult used for transcriptome sequencing.

Sequence data preprocessing and assembly

The raw sequence reads corresponding to the two tissue types were error corrected using the software BLESS (Heo et al., 2014) version 0.17 (https://goo.gl/YHxlzI, https://goo.gl/vBh7Pg). The error-corrected sequence reads were adapter and quality trimmed following recommendations from MacManes (2014) and Mbandi (Mbandi et al., 2014). Specifically, adapter sequence contamination and low quality nucleotides (defined as Phred < 2) were removed using the program Trimmomatic version 0.32 (Bolger, Lohse & Usadel, 2014) called from within the Trinity assembler version 2.1.1 (Haas et al., 2013). Reads from each tissue were assembled together to created a joint assembly of adult and larva transcripts using a Linux workstation with 64 cores and 1 Tb RAM. We used flags to indicate the stranded nature of sequencing reads and set the maximum allowable physical distance between read pairs to 999nt (https://goo.gl/ZYP08M).

The quality of the assembly was evaluated using transrate version 1.01 (Smith-Unna et al., 2016; https://goo.gl/RpdQSU). Transrate generates quality statistics based on a process involving mapping sequence reads back to the assembled transcripts. Transcripts supported by properly mapped reads of a sufficient depth (amongst other things) are judged to be of high quality. In addition to generating quality metrics, transrate produces an alternative assembly with poorly-supported transcripts removed. This improved assembly was used for all downstream analyses and QC procedures. We then evaluated transcriptome completeness via use of the software package BUSCO version 1.1b (Simão et al., 2015). BUSCO searches against a database of highly-conserved single-copy genes in Arthropoda (https://goo.gl/bhTNdr). High quality, complete transcriptomes are hypothesized to contain the vast majority of these conserved genes as they are present in most other species.

To remove assembly artifacts remaining after transrate optimization, we estimated transcript abundance using 2 software packages—Salmon version 0.51 (Patro, Suggal & Kingsford, 2015; https://goo.gl/01UIF6) and Kallisto version 0.42.4 (Bray et al., 2015; https://goo.gl/BsQMpr). Transcripts whose abundance exceeded 0.5 TPM in either adult or larval datasets using either estimation method were retained. TPM is different from FPKM with regards to the order of operations performed, as TPM normalizes for gene length first. We evaluated transcriptome completeness and quality, again, after TPM filtration, using BUSCO and transrate, to ensure that our filtration processes did not significantly effect the biological content of the assembly.

We identified and removed potential plant, fungal, bacterial and vertebrate contamination by using a blastx search. We created a custom protein database based on the collection of protein sequences from each taxonomic group available for download from RefSeq (ftp://ftp.ncbi.nlm.nih.gov/refseq/release/). We queried this database, inferring a given sequence to be a contaminate if the best blast (=lowest e-value) was plant, fungal, bacterial or vertebrate on origin, rather than invertebrate.

Assembled sequence annotation

The filtered assemblies were annotated using the software package dammit (https://github.com/camillescott/dammit; https://goo.gl/05MY5i). Dammit coordinates the annotation process, which involves use of blast (Camacho et al., 2009), TransDecoder (version 2.0.1, http://transdecoder.github.io/), and hmmer version 3.1b1 (Wheeler & Eddy, 2013). In addition to this, putative secretory proteins were identified using the software signalP, version 4.1c (https://goo.gl/FaOQSj).

In addition to this, we attempted to identify loci involved in color, color patterning, and more generally phenotypic polymorphism. To accomplish this, we downloaded a set of 1,008 candidate genes previously identified as underlying phenotypic evolution in other species, from Dryad (Martin & Orgogozo, 2013a; Martin & Orgogozo, 2013b). We used the gene name listed in this dataset to download the protein sequence from Uniref90. We used a blastX search strategy to identify potential homologues in the dataset.

To identify patterns of gene expression unique to each life stage, we used the expression data as per above. We identified transcripts expressed in one stage but not the other, and cases where expression occurred in both life stages. The Uniprot ID was identified for each of these transcripts using a blastX search (https://goo.gl/J9saMj), and these terms were used in the web interface Amigo (Carbon et al., 2009) to identify Gene Ontology terms that were enriched in either adult or larva relative to the background patterns of expression. The number of unique genes contained in the joint assembly was calculated via a BLAST search of the complete gene sets of human Homo sapiens, fruit fly, Drosophila melanogaster, and the flour beetle, Tribolium casteneda.

Results and Discussion

Data availability

All read data are available under ENA accession number PRJEB13023. Assemblies and data matrices are available at https://goo.gl/D3xh65, and will be moved to Dryad following manuscript acceptance.

RNA extraction, assembly and evaluation

RNA was extracted from whole bodies of both the adult and the larva stage of a single Harmonia axyridis. The quality was verified using a Tapestation 2200 (all RIN scores >8) as well as a Qubit. The initial concentration for the larva sample was 83.2 ng*µL −1, while the initial concentration for the adult sample was 74.7 ng*µL −1. The number of strand-specific paired end reads contained in the adult and larva libraries were 58 million and 67 million, respectively. The reads were 125 base pairs in length.

The raw Trinity assembly of the larval and adult reads resulted in a total of 171,117 contigs (82 Mb) exceeding 200nt in length. 526 contaminate sequences were removed. This assembly was evaluated using Transrate, producing an initial score of 0.10543, and optimized score of 0.29729. The optimized score indicated that the optimized assembly was better than 50% of NCBI-published de novo transcriptomes (Smith-Unna et al., 2016). This transrate optimized assembly (89,305 transcripts, 62 Mb) was further filtered by removing transcripts whose expression was less than 0.5 TPM. After filtration, 33,648 transcripts (44 Mb) remained. To assess for the inadvertent loss of valid transcripts, we ran BUSCO before and after this filtration procedure. The percent of Arthropoda BUSCO’s missing from the assembly rose slightly, from 18% to 21%. Transrate was run once again, and resulted in a final assembly score of 0.29112. This score is indicative of a high-quality transcriptome appropriate for further study (Smith-Unna et al., 2016). In an attempt at understanding how many distinct genes our transcriptome contained, we conducted a blast search against Homo sapiens, Drosophila melanogaster, and Tribolium casteneda. This search resulted in 7,246, 7,739, and 7,741 matches unique to Harmonia axyridis, which serve as estimates of the number of unique genes expressed in these two life stages. The final assembly is available at https://goo.gl/nWdBuv.

Annotation

The assembled transcripts were annotated using the software package dammit!, which provided annotations for 23,304, or 69% of the transcripts (available here: https://goo.gl/gpGXLG). These annotations included putative protein and nucleotide matches, 5- and 3-prime UTRs, as well as start and stop codons. In addition to this, analysis with Transdecoder yielded 14,518 putative protein sequences (available here: https://goo.gl/qVLWwD), which were annotated by 4,139 distinct Pfam protein families, while 176 transcripts were determined to be non-coding (ncRNA) based on significant matches to the Rfam database (available here: https://goo.gl/x1n7jC). Lastly, 2,925 proteins (7.8% of total) were determined to be secretory in nature by the software package signalP (available here: https://goo.gl/z0ra1g).

Annotation of the sequence dataset resulted in the identification of host of transcripts that may be of interest to other researchers including: 43 heat-shock and 8 cold-shock transcripts, 87 homoebox-domain containing transcripts, 122 7-transmembrane-containing (18 GPCR’s) transcripts, 13 solute carriers, 143 ABC-transport-containing transcripts, and 21 OD-S (pheromone-binding) transcripts.

A complement of immune-related genes were discovered as well. These include a single member of the Attacins and Coleoptericins, two TLR-like genes, seven Group 1, and 34 Group 2 C-type lectin Receptors (CLRs). Two CARD-containing Cytoplasmic pattern recognition receptor (CRR) genes were discovered, as were 3 MAP kinase containing transcripts. Finally, 119 RIG-I-like receptors (RLR) were found.

The focused search for genes previously implicated in phenotypic polymorphism (Martin & Orgogozo, 2013a; Martin & Orgogozo, 2013b) identified 483 Harmonia transcripts, corresponding to 65 distinct loci (Table S1). These loci include the transcription factors bab1, Distal-less, Optix, enzymes BCMO1, ebony, yellow, and transporters ABCC2, SLC24A5, and TPCN2, all related to color and color patterning. These genes, and the others identified in Table S1 to are likely to provide fodder for future research.

Gene expression was estimated for each transcript for both adult and larva (Fig. 2, data available here: https://goo.gl/wM3TV7). For all transcripts expressed in the adult, the mean TPM = 29.7 (max = 80,016.9, SD = 500.8) while the mean larval TPM = 29.71 (max = 166,264, SD = 941). When analyzing transcripts found uniquely in these two tissues, mean adult TMP = 9.9 (max = 3,037, SD = 86) and for larva TPM = 8.6 (max = 687, SD = 41).

Figure 2 The Venn diagram representing the number of transcripts expressed in both adult and larva, as well as those expressed uniquely in one or the other.

Analysis of the differences between adult and larval life stages were carried out as well. Because these life stages were only sequenced with a single individual each, they should be interpreted with some caution. The vast majority of transcripts were observed in both life stages (n = 30, 630, 91%), with a small number being expressed uniquely in larva (n = 1, 094) and adult (n = 1, 922). Of these transcripts expressed uniquely in either larva or adult, 45% and 42%c were annotated using at least one method via the software package dammit. 6.1% and 4.6%, respectively, were found to be secretory in nature via signalP analysis.

Conclusions

Phenotypic polymorphisms and striking visual phenotypes have fascinated scientists for many years. The breadth of evolutionary causes for the maintenance of these phenotypes are as numerous as the species that display them. One organism, Harmonia axyridis, provides a unique opportunity to explore the genetic basis behind the maintenance of an easy to quantify variation—elytral spot number. While understanding these genomic mechanisms is beyond the scope of this paper, we do provide a reference transcriptome for H. axyridis, a foundational resource for this work.

This study indicates that most gene expressed at levels greater than .5 TPM were shared seen in both the adult and larval individuals. While the majority of proteins identified in the assembled transcriptome were structural in function, analyses of protein families using the Pfam database indicated the presence of pigment proteins. In particular, RPE65, which functions in the cleavage of carotenoids, was found. In H. axyridis, increased carotenoid pigmentation has been linked to increased alkaloid amounts (Britton, Liassen-Jesen & Pfander, 2008). In addition, the elytral coloration of the seven spot ladybug, Coccinella septempunctata, is a result of several carotenoids (Britton, Liassen-Jesen & Pfander, 2008). While larva are mostly black (Fig. 1A), we posit that the orange sections on the lower back could be due to carotenoid production. Moreover, this study provides a necessary foundation for the continued study of the genetic link between genes and the maintenance of variation in H. axyridis.

Supplemental Information

Table S1 Genes previously implicated in phenotypic polymorphism (Martin & Orgogozo, 2013a; Martin & Orgogozo, 2013b) that are present in the Harmonia axyridis transcriptome

Click here for additional data file.

Additional Information and Declarations

Competing Interests

Author Contributions

Data Availability

The authors declare there are no competing interests.

Lindsay A. Havens conceived and designed the experiments, performed the experiments, analyzed the data, wrote the paper, prepared figures and/or tables, reviewed drafts of the paper.

Matthew D. MacManes conceived and designed the experiments, performed the experiments, analyzed the data, contributed reagents/materials/analysis tools, wrote the paper, prepared figures and/or tables, reviewed drafts of the paper.

The following information was supplied regarding data availability:

GitHub: https://github.com/macmanes-lab/harmonia_manuscript/blob/master/data.md#reads.

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
