# Peer review of "Characterizing the adult and larval transcriptome of the multicolored Asian lady beetle, Harmonia axyridis"

_PeerJ, doi:10.7717/peerj.2098_

## Round 0.1 · original submission · Major Revisions

· Academic Editor

Major Revisions

As you will see the reviewers agree on the merit of the publication and the care taken when describing the methods. Ive selected major revisions as i think this requires time/effort to address concerns. However i dont think that this should be seen as a criticism of the manuscript just a request to make the paper as good as as can be. I look forward to your response.

·

Basic reporting

Dear Authors and PeerJ Editor,

I am not an expert in studying phenotypic polymorphism in insects or any other organism. Therefore, I cannot comment on the accuracy of the introduction. However, as a lay reader, I agree that the background and motivation for this study in the introduction section are clear and make sense.

The research aim is clearly stated "To begin to understand the genomics of elytral coloration and spot patterning, we sequenced the transcriptome of an late-stage larva and adult ladybug", and the structure of the article is acceptable as a transcriptome assembly report. Figures are acceptable (although one more is requested, see section on General comments to authors).

All assembled data / annotations have been made available as amazon S3 links but should be on FigShare/DataDryad type sites before publication. Raw reads must be in ENA/SRA before publication.

Experimental design

A well produced reference transcriptome is much appreciated. This paper uses the latest tools to get around the problems of [a] excessive transcripts (as produced by trinity), and [b] low quality transcripts (possibly transcriptional noise).

This paper also has the opportunity to become a very useful reference on 'how to do an illumina rna-seq transcriptome assembly' as it is well described, and comes with excellent online methods. Thank you for providing that. Just a suggestion - consider archiving the github content on figshare (figshare allows archiving github releases I believe) because it is easy for a github repo to accidentally delete/change content and then not many people will be able to track the exact documents linked with this paper.

My only methodological concern is that there were no checks done for any contamination, especially as sequencing was done from wild-caught samples. I could be wrong, but the RNAseq protocol does not seem to be PolyA selected, and therefore could include bacterial contaminants? I found a few examples (eg Harmonia_transcript_905, Harmonia_transcript_1494, Harmonia_transcript_1780) that may be more likely to be bacterial sequences, and even some possibly plant sequences (Harmonia_transcript_8315, Harmonia_transcript_21722,21723,21724). It's possible that these are real Harmonia sequences but I'd like a little cross checking done to be sure.

There are some specific comments about cutoffs / parameters that I list in the general comments section.

Validity of the findings

PeerJ guidelines clearly state that "Decisions are not made based on any subjective determination of impact, degree of advance, novelty, being of interest to only a niche audience, etc" and so I have not considered any of these factors.

Even given the limited scope of the study, I think a slightly more detailed results and conclusion section is needed. If the goal is to establish or provide resources for Harmonia as a model for phenotypic variance, then a more systematic search for homologues of known colour/patterning genes would be better than simply reporting one protein RPE65. Ideally, there should be lists of homologues of the known relevant proteins to make it easier for other researchers to work with those genes.

Additional comments

Some comments on individual lines in the manuscript:

55. "while offering several key benefits". Just to clarify - are the key benefits the ones described in the next few paragraphs? Or are there other benefits as well?

91. TruSeq stranded RNA prep kit - is this polyA selected? I don't know enough about lab kits and techniques, but a quick look at http://www.illumina.com/products/truseq_stranded_total_rna_library_prep_kit.html suggests that there are multiple kits with this kind of name so which one was used? If it is not PolyA selected, then it may accidentally pick up RNA from organisms other than the target organism. Even if it is PolyA selected, RNA from plant material can get picked up (see my earlier comment on contamination checks).

115. I'm curious about peak memory usage (not essential for paper, but would be a useful data point for others looking to replicate)

117. Is there a reason for 999nt? It seems excessive given that RNA libraries are often not size selected at that size.

127. Delete first 'contain'

159. Needs the raw trinity assembly, and the filtered ones (to be considered a complete data record). Raw reads should be in ENA/SRA. All other datasets (assemblies/annotations/expression levels etc) should be on figshare/datadryad before publication, not on Amazon S3

170. Are these good scores? Bad scores? Citation needed. I checked how transrate works to give an optimised assembly at http://hibberdlab.com/transrate/metrics.html but your paper should cite that or explain it.

171. 0.5 TPM seems arbitrary. Some people seem to plot a histogram of expr levels and use that to determine cutoff.

172. 40 Mb should be 44 Mb (43.958...)

175. How do we know this? Is there a citation? Or an explanation of this score?

177-179. Awkwardly worded. "Search resulted in ... unique matches... which serve as estimates of unique genes expressed..." - unique to this species? I think you mean unique as in not duplicated or not-alternatively-transcribed, but that's not at all clear from this sentence.

206: TMP - change to TPM

205-209. Figure 2 is fine. But rather than reporting means, I think a figure showing histograms of expression (log values ok, TPM bins of ... 0.01 0.1 1 10 100 1000 etc) would be helpful, and would also help justify 0.5 TMP cutoff. Also which software gave these results? Salmon or Kallisto?

Best wishes,

- Sujai

Reviewer 2 ·

Basic reporting

This paper is a first step to address phenotypic polymorphism by using the multicoloured Asian lady beetle and assess it's gene expression with NGS based transcriptomics. The problem of phenotypic polymorphism is laid out in the introduction and the unresolved scientific question is postulated. The methods and results are described in a cursory “Results & Discussion” section that follows. The main flaw in this paper is that the introduction sets up a interesting question, but the analysis fails to even offer any possibilities of any gene sets that might be the drivers of the phenotypic polymorphisms. This lack of analytical insight leaves the reader with a sense of disappointment, more than anything and this reviewer would like to see some attempt at addressing the scientific question(s) initially outlined in the introduction.

Experimental design

Please include the common names of organisms (if they have one) when describing them in the text – not everyone knows what is the scientific name of every given organism. At the end of page 2 (line 66) does this not have serious implications for climate change? Maybe a reference for this might be appropriate and could potentially expand the audience for this type of work – the Asian lady beetles might be harbingers of climatic changes. The Venn diagram is too big, and the transcripts that are included seem to be both the annotated and non-annotated. Of the up/down group, how many are annotated and how many are non-annotated? The emergence of novel transcripts might be worth listing separately (if they translate without truncation – i.e. they aren't in silico assembly artifacts). This is a more conservative approach, but might be more realistic as Trinity can be known to overestimate the number of transcripts. Additionally, transcripts less than about 400 nt sometimes are complete garbage – but it seems that the use of a filtre to remove those that had a hit of less than 0.5 TPM has adjusted for this issue. A brief comment on the usage of TPM, versus FPKM (old RPKM) might be warranted – many readers will not understand the difference, and more importantly, many will not understand the advantage of using TPM for comparisons between samples when using RNA-Seq data with different read depths.
An overall general concern that this reviewer has with the Results/Discussion, or interpretation, is that there has been little effort to parse the identified DE transcripts. How many transcripts can the authors attribute to the difference between life stages versus the differences between the phenotypic plasticity versus the complexity of the physiological and/or morphological differences? This seems to be a major outstanding question, yet no effort seems to have been made to sort one from the other in terms of coordinated expression. The only binning of the data are the general categories outlined from lines 184 ~ 209. What about all the microRNAs? There's a lot more that can said about this data than I think was discussed in this manuscript without overreaching the importance or impact of this study.
Figure 2 (as with the vast majority of all the URLs) is unresolvable (in other words, the links in your manuscript don't work!). This is a major oversight on the part of the authors and is a fatal flaw in this manuscript submission. The only URL that worked took me to this page: https://github.com/macmanes-lab/harmonia_manuscript/blob/master/data.md . As you'll note, there is no “Figure 2” - or at least it wasn't obvious to this reviewer where I could find it. In any case, it seems Fig 2 is a pretty big deal, why would the authors not include it in the main body of the manuscript?

Validity of the findings

Specific Concerns (Major Issues):

(1) How long was there between immersion in the RNAlater and freezing? If it was immediate, there would be degradation.

(2) Was the TruSeq kit the V2? If so, on line 92, do you mean to say “unique index code”?

(3) How were you able to pool the samples in an equimolar fashion? Using the KAPA qPCR kit? If simply by Qubit and/or Tapestation, then your clustering could have been significantly biased between adult and larva. Quantitaion by KAPA qPCR identifies the percentage of your completed library that can be clustered for PE sequencing.

(4) Figure 1b isn't very orange – if you call something orange, it should be orange.

(5) This last point is my major grip with the entire manuscript: Why did you use just one individual per life stage? This could be (may be) a catastrophic decision – you might have (by chance) sequenced a weirdo, or a sick individual, or something else. Maybe a bad isolation? RNA degradation? There are potentially so many reasons why only using one technical replicate from one biological replicate per life stage is problematic. I don't know how to address this short coming.

Specific Concerns (Minor Issues):

(6) Line 164: Change <ng/uL> to <ng * µL-1> – and all other instances of this type of misformatting.

(7) Line 206 & 208: TMP should be TPM?

(8) Lines 205 ~ 209 inclusive. I have no idea what any of this means – at all. Maybe you can make some biological inferences as opposed to giving the reader mean values of a subjective metric?

(9) Lines 218 ~ 220: There must be more to this study than “secretory in nature”.

(10) Line 127/128: This last sentence doesn't make sense, can you elaborate as to why?

Additional comments

Overall, I like the study, but this manuscript isn't fully baked. You need (and should) spend a little more time with the analysis and try to tell the reader a story that relates to your initial questions based on your findings.

Reviewer 3 ·

Basic reporting

The manuscript is written well and seems to conform to the journal standard. Given the visual nature of the underlying motivation the photos of the sequenced individuals was a nice touch.

My only concern is raw data should be submitted to long-term repositories like SRA. From the links it appears both raw and processed data are on S3 @Amazon. It is unclear in the manuscript if this storage will be maintained indefinitely, while SRA will be maintained per NCBI's mandate.

Experimental design

The informatics seems well done and sufficiently described. My only concern is if there were known endosymbiants / bacteria in these individuals that could show up in the transcript number and inflate the "unique" transcript count.

Similarly, it would have been interesting to look for contaminants in the ~1000 "uniquely" expressed genes by looking at their annotations. A small % are secretory, but are any important? I agree that not much can be said here given there was only one individual sequenced -- so almost zero statistical power -- but it is something the authors discuss.

Validity of the findings

See earlier comments re: submission to SRA and the section on unique expressed genes that may be contaminants. With the filters applied I do not think there will be much contamination, but it is possible.

Overall a simple but well done gene discovery exercise in this ladybug.

Additional comments

My only real issue with the text is the portion in the conclusions where you state "... most gene expression profiles are shared across life stages of H. axyridis." Since really no differential expression can be assessed here with single individuals, "profile" seems strong. Further, you only looked at two of the four stages. What if expression is radically different in the pupal stage?

A more accurate statement given the data is "most genes expressed at levels greater than 0.5 TPM were expressed in both the larval and adult individual." The clarification is the expression filter may favor "housekeeping" genes that should be expressed in multiple stages.

Finally, thought this was a cool paper. There was an "invasion" of our campus when I was a graduate student in the midwest so I'm familiar with the species. PeerJ seems to be a good place to announce the data now available.

---

## Round 0.2 · accepted · Accept

· Academic Editor

Accept

Thanks for addressing concerns of the reviewers. A really nice paper!